# Properly Substituted Cyclic Bis-(2-bromobenzylidene) Compounds Behaved as Dual p300/CARM1 Inhibitors and Induced Apoptosis in Cancer Cells

**DOI:** 10.3390/molecules25143122

**Published:** 2020-07-08

**Authors:** Rossella Fioravanti, Stefano Tomassi, Elisabetta Di Bello, Annalisa Romanelli, Andrea Maria Plateroti, Rosaria Benedetti, Mariarosaria Conte, Ettore Novellino, Lucia Altucci, Sergio Valente, Antonello Mai

**Affiliations:** 1Dipartimento di Chimica e Tecnologie del Farmaco, ‘Sapienza’ Università di Roma, 00185 Roma, Italy; rossella.fioravanti@uniroma1.it (R.F.); elisabetta.dibello@uniroma1.it (E.D.B.); annalisa.romanelli@uniroma1.it (A.R.); 2Dipartimento di Farmacia, Università di Napoli ‘Federico II’, 80131 Napoli, Italy; stefano.tomassi@unina.it (S.T.); ettore.novellino@unina.it (E.N.); 3Dipartimento di Neuroscienze, Salute Mentale e Organi di Senso–Nesmos, ‘Sapienza’ Università di Roma, 00185 Roma, Italy; andrea.plateroti@gmail.com; 4Dipartimento di Medicina di Precisione, Università degli Studi della Campania Luigi Vanvitelli, 80138 Napoli, Italy; rosaria.benedetti@unicampania.it (R.B.); mariarosaria.conte@unicampania.it (M.C.); lucia.altucci@unicampania.it (L.A.)

**Keywords:** epigenetics, histone acetylation, histone methylation, drug discovery, multi-target agents

## Abstract

Bis-(3-bromo-4-hydroxy)benzylidene cyclic compounds have been reported by us as epigenetic multiple ligands, but different substitutions at the two wings provided analogues with selective inhibition. Since the 1-benzyl-3,5-bis((*E*)-3-bromobenzylidene)piperidin-4-one **3** displayed dual p300/EZH2 inhibition joined to cancer-selective cell death in a panel of tumor cells and in in vivo xenograft models, we prepared a series of bis((*E*)-2-bromobenzylidene) cyclic compounds **4a**–**n** to test in biochemical (p300, PCAF, SIRT1/2, EZH2, and CARM1) and cellular (NB4, U937, MCF-7, SH-SY5Y) assays. The majority of **4a**–**n** exhibited potent dual p300 and CARM1 inhibition, sometimes reaching the submicromolar level, and induction of apoptosis mainly in the tested leukemia cell lines. The most effective compounds in both enzyme and cellular assays carried a 4-piperidone moiety and a methyl (**4d**), benzyl (**4e**), or acyl (**4k**–**m**) substituent at N1 position. Elongation of the benzyl portion to 2-phenylethyl (**4f**) and 3-phenylpropyl (**4g**) decreased the potency of compounds at both the enzymatic and cellular levels, but the activity was promptly restored by introduction of a ketone group into the phenylalkyl substituent (**4h**–**j**). Western blot analyses performed in NB4 and MCF-7 cells on selected compounds confirmed their inhibition of p300 and CARM1 through decrease of the levels of acetyl-H3 and acetyl-H4, marks for p300 inhibition, and of H3R17me2, mark for CARM1 inhibition.

## 1. Introduction

Polypharmacology is a new emerging approach for chemotherapy, particularly useful when the single-targeted therapy tailored for a specific disease remains without or loses its effect [1]. This can happen because multifactorial diseases, such as neurodegenerative disorders or cancer, typically feature multiple dysfunctions of several biological pathways, which can elicit resistance towards a single target modulator through their robustness and redundancy [2]. Multi-targeting compounds differ from promiscuous drugs because the first are directed towards different targets and pathways all related to the same pathology, whereas the latter possess a broad spectrum of biological activities directed against targets unrelated to the same disease, and are joined to a plethora of adverse reactions due to the presence of off-target effects [3]. Particularly in cancer, the contribution of genetic, epigenetic, and metabolic dysfunctions may sustain the insurgence and the development of the disease, thus its treatment could likely require complex therapeutic approaches, hijacking different signal pathways and cross-talks, all related to epigenetic and not epigenetic phenomena [4].

Transcriptional dysregulations and epigenetic dysfunctions are common and recurring themes in cancer. Alterations in histone acetylation and/or methylation or in DNA methylation modify the transcription regulation leading to aberrant events characterized by either gene silencing or activation [5,6,7,8,9]. Then, such dysfunctions can activate selected cellular programs for cell cycle arrest and block of differentiation and/or apoptosis allowing cancer initiation and development, and can be managed by using orthosteric and allosteric inhibitors, peptidomimetics, and degraders [10].

In 2008, pursuing our studies on design, synthesis, and biological evaluation of small molecule modulators of epigenetic targets as anticancer agents, we prepared a large series of compounds with general formula **1** (Figure 1). These structures hold two bromo- or dibromo-hydroxyphenyl portions separated by spacers with different length and size, based on the similarity with some dyes previously reported as histone methyltransferase inhibitors (HMTs) in a pioneering study [11]. We tested these compounds against HMTs (belonging to both PRMT (protein arginine methyltransferase) and KMT (histone lysine methyltransferase) families) and, due to their strict similarities with some described histone acetyltransferase (HAT) or sirtuin modulators, against p300 HAT and SIRT enzymes [12,13]. Among such compounds, those carrying two 3,5-dibromo-4-hydroxyphenyl/benzylidene moieties linked through a penta-1,4-dien-3-one, (hetero)cycloalkanone, or 1,1-(1,3-phenylene)diprop-2-en-1-one spacer behaved as epigenetic multiple ligands (epi-MLs), capable to inhibit at the same time all the tested enzymes [13]. When tested on the U937 human acute monocytic leukemia (AML) cell line, some epi-MLs induced high apoptosis levels and/or massive, dose-dependent cytodifferentiation, whereas the related single-target inhibitors were ineffective or showed a weak effect [13]. A combination of distinct substitution patterns on the two phenyl/benzylidene wings together with an appropriate spacer provided compounds with a better spectrum of activities [12,13,14,15,16]. The insertion of the 3-bromo-4-hydroxybenzylidene portions at the 3,5 positions of the 1-benzylpiperidin-4-one yielded **2** (Figure 1), inactive against PRMT1, SET7, and p300 HAT, and exhibiting selective inhibition of CARM1 [17] and EZH2 (unpublished results), a KMT highly involved in cancer diseases. The change of 3-bromo-4-hydroxy- into 3-bromo-4-methoxyphenyl/benzylidene group applied to the penta-1,4-dien-3-one or cyclohexan-1-one scaffold led to loss of inhibition towards PRMT1 and CARM1, and gain of activity against SETD8 and, to a lesser extent, EZH2 in a panel of five tested KMTs (SET7, SETD8, G9a, SET7/9, and EZH2) [14,18,19,20]. The removal of the 4-hydroxy substituents from **2** furnished MC2884 (**3**, Figure 1), a dual p300/EZH2 inhibitor able to induce cancer-selective cell death in both solid and blood cancers in vitro, ex vivo, and in vivo xenograft models [21,22].

The considerable anticancer activity observed with **3** prompted us to explore the biochemical and biological effects of the shift of the bromine substituents from the 3- to the 2-benzylidene positions, using the cyclohexanone, tetrahydro-4*H*-pyran-4-one or piperidin-4-one moiety as connection units (compounds **4a**–**n**, Figure 1). At the piperidone N1 position, besides hydrogen we inserted a methyl, benzyl, 2-phenethyl, 3-phenylpropyl, 2-oxo-2-phenylethyl, 3-oxo-3-phenylpropyl, 4-oxo-4-phenylbutyl, benzoyl, 2-phenylacetyl, 3-phenylpropanoyl, or 3-phenylacryloyl substituent. Compounds **4a**–**n** have been evaluated in p300, PCAF, SIRT1/2, EZH2 and CARM1 enzyme assays, and in NB4 human acute promyelocytic leukemia (APL), U937 AML, MCF-7 breast cancer and SH-SY5Y neuroblastoma cells, to determine their effects in cell cycle phases and apoptosis (sub-G1 peaks) induction. Western blot analyses of the levels of acetyl-H3, acetyl-H4, acetyl-H3K9/14, and acetyl-α-tubulin, marks of p300 activity in cells, as well as of those of H3K27me3 (trimethylated histone 3 lysine 27) and H3R17me2 (dimethylated histone 3 arginine 17), histone marks of cellular activity of EZH2, and CARM1, respectively, have been performed for selected derivatives in NB4 and MCF-7 cells to correlate the observed biological effects with the modulation of the cited epigenetic targets.

## 2. Results and Discussion

### 2.1. Chemistry

The known 2,6-bis((*E*)-2-bromobenzylidene)cyclohexan-1-one **4a** [23], 3,5-bis((*E*)-2-bromobenzylidene)tetrahydro-4*H*-pyran-4-one **4b** [24], 3,5-bis((*E*)-2-bromobenzylidene)piperidin-4-one **4c** [25], and 3,5-bis((*E*)-2-bromobenzylidene)-1-methylpiperidin-4-one **4d** [24,25] were prepared through aldol condensation between 2-bromobenzaldehyde and the appropriate 6-membered cyclic ketones by replacing sodium hydroxide, used as the basic catalyst in the previous reports [23,24,25], with barium hydroxide and methanol for 2 h at room temperature. Starting from **4c**, compounds **4e**–**j** were obtained by alkylation with suitable alkyl chlorides heating at 60 °C for 2 h in the presence of K_2_CO_3_ in dry acetonitrile (MeCN) (**4e**–**i**), or by one-pot reductive amination with 4-oxo-4-phenylbutanal and sodium triacetoxyborohydride [NaBH(AcO_3_)_3_] in dry dichloromethane (DCM) (**4j**) for 2 h at room temperature (Scheme 1). Compounds **4k**–**n** were prepared by acylating the piperidin-4-one N1 nitrogen with different acyl chlorides, characterized by increasing distance between the carbonyl group and the phenyl ring, in the presence of triethylamine (TEA) in dry DCM for 1 h at room temperature (Scheme 1). 

### 2.2. p300, PCAF, SIRT1/2, EZH2, and CARM1 Enzyme Assays

The newly synthesized compounds **4a**–**n** were tested to determine their effects against acetylation and methylation epigenetic targets. First, **4a**–**n** were screened in vitro against the two HAT enzymes p300 and PCAF. The IC_50_ values, when possible, were determined in a 10-dose mode, in triplicate, with 2-fold serial dilution starting from 200 μM solutions. In these assays, histone H3 and [acetyl-^3^H]-acetyl-CoA were used as substrate and co-substrate, respectively, and C-646 (for p300) and anacardic acid (for PCAF) were added as reference compounds. To determine the activity against SIRT1 and -2, **4a**–**n** were tested starting from 200 μM solutions using a modified Fluor de Lys fluorescence-based assay with amino acids 379–382 (Arg-His-Lys-Lys(Ac)) of human p53 conjugated with aminomethylcoumarin as substrate and NAD^+^ as co-substrate. The SIRT1 inhibitor selisistat was used as internal reference compound. Furthermore, **4a**–**n** were tested against a human five-membered Polycomb repressive complex 2 (PRC2) containing EZH2, embryonic ectoderm development (EED), suppressor of zeste 12 (SUZ12), RbAp48, and adipocyte enhancer-binding protein (AEBP2), to assess their capability to inhibit the catalytic unit EZH2. According to a 2-fold serial dilution pattern, **4a**–**n** were tested in a 10-dose IC_50_ evaluation in duplicate starting from 400 μM as initial concentration. The core histone and ^3^H-SAM were used as substrate and co-substrate, respectively, whereas *S*-adenosyl-l-homocysteine (SAH) was used as reference compound. Finally, **4a**–**n** were screened against CARM1 in agreement to the experimental protocol already reported above for p300 and PCAF, but using histone H3 and ^3^H-SAM as substrate and co-substrate, respectively, and SAH as the internal reference. 

When tested toward histone (de)acetylation enzymes, the majority of **4a**–**n** demonstrated to consistently inhibit the HAT p300, with some compounds displaying a submicromolar activity. Differently, **4a**–**n** were completely ineffective against PCAF as well as SIRT1/2 (Table 1). In particular, **4a**, **4b**, **4d**, and **4e**, bearing the cyclohexanone, the tetrahydro-4*H*-4-pyranone and the *N*-methyl- and *N*-benzyl-4-piperidone spacer, exhibited single-digit micromolar IC_50_ values against p300 activity. The elongation of the benzyl substituent at the N1 4-piperidone position (see compounds **4f** and **4g**) significantly decreased the acetyltransferase inhibition, that was promptly restored when a carbonyl function was added to the N1-alkylphenyl chain (**4h**–**j**, stabilized on single-digit micromolar values). The replacement of the alkylphenyl group at the N1 4-piperidone position with an acyl moiety, such as in **4k**–**n**, gave a gain of potency against p300 in particular with **4l** and **4m** (IC_50_ values in the submicromolar range).

Towards PRC2/EZH2, **4a**–**n** generally displayed low, if any, inhibition (Table 1). Only N1-substituted 4-piperidone compounds exhibited some activity, with the N1-acyl derivatives **4l** and **4m** being the most effective. The *N*-methyl and *N*-benzyl analogues **4d** and **4e** showed low EZH2 inhibition, which was lost with **4f** and **4g** (carrying longer substituents at N1) and partially restored by the introduction of the ketone function in the alkylaryl chain.

Against CARM1, the N1-substituted 4-piperidone compounds displayed the highest activity, reaching in some cases IC_50_ values at low micromolar/submicromolar levels (Table 1). The insertion of the methyl or benzyl group at N1 furnished quite potent compounds (**4d** and **4e**), while 2-phenylethyl (**4f**) and 3-phenylpropyl (**4g**) substitution were less effective. Single-digit micromolar IC_50_ values were shown by compounds bearing a carbonyl group (ketone or amide function) in the N1-side chain, while **4l** and **4m** were the most potent (IC_50_ values at submicromolar level).

### 2.3. Biological Evaluation in Cancer Cells

Compounds **4a**–**n** were tested at 5 μM for 30 h in NB4 PML and U937 AML cells among hematological cancers, and in MCF-7 breast cancer and SH-SY5Y neuroblastoma cells among solid cancers, to determine their effects in cell cycle and apoptosis induction. In NB4 cells, the majority of compounds (**4a**–**c**, **4g**–**k**) gave a block in G2/M phase, while in U937 cells only **4f**, **4l**, and **4m** displayed the same effect, and **4g** and **4n** arrested the cell cycle at the G1/S phase (Figure 2A,B). In MCF-7 cells, most compounds exhibited a profound alteration of the cell cycle with a block in the G1/S phase and the almost total absence of cells either in the G2/M phase (**4a**, **4c**, **4d**, **4e**, **4j**, **4m**, and **4n**) or in the S phase (**4f**–**i**, **4k**, **4l**, and **4n**) (Figure 2C). SH-SY5Y cells treated with **4a**–**n** showed moderate alteration of the cell cycle, with **4h**, **4j**, **4l**, **4m** and **4d**, **4f**, **4k** displaying an arrest in the G1/S phase or in the G2/M phase, respectively (Figure 2D).

In the same cell lines, the induction of apoptosis has been evaluated through the percentage of the sub-G1 peaks. In NB4 cells, **4l** and **4m** displayed the highest induction (% sub-G1 peaks = 67 and 56, respectively) (Figure 3A), demonstrating the crucial role of the insertion of the phenylacetyl (**4l**) and 3-phenylpropionyl (**4m**) moiety at the N1 4-piperidone position to elicit this cellular effect. It is noteworthy that **4l** and **4m** are also the most potent among the described dual p300/CARM1 inhibitors, reaching IC_50_ values at submicromolar level in both the enzymatic assays (Table 1). In the same cell line, **4d**, **4e**, **4h**, and **4k** showed from 30 to 40% of sub-G1 peaks. In U937 cells, **4e**, **4k**, **4l**, and **4m** exhibited a percentage of apoptosis over 30% (Figure 3B), the others being less effective. In the tested solid tumor cell lines, MCF-7 and SH-SY5Y, the **4** compounds were generally less potent in inducing apoptosis respect to blood cancers: In MCF-7 cells, only the N1-acyl derivatives **4k**–**m** exceeded 20% of sub-G1 peaks (Figure 3C), and in SH-SY5Y cells **4m** was the most potent with a percentage of apoptosis induction of 17% (Figure 3D).

The most potent derivatives as apoptosis inducers, **4l** and **4m**, were tested in mesenchymal progenitor MePR2B cells, an immortalized non-cancer cell system [26,27], to determine their differential toxicity. After treatment at 5 μM for 30 h, **4l** and **4m** reduced the percentage of cell viability to 61.2 ± 0.8 and 85.5 ± 3.7, respectively, in comparison with the control (100%).

Western blot analyses were carried out on NB4 and MCF-7 cells treated with a subset of compounds **4** (5 μM, 30 h), to confirm that the observed effects are due to modulation of p300, CARM1, and eventually EZH2 activities. Suberoylanilide hydroxamic acid (SAHA), a FDA-approved HDAC inhibitor [28], and GSK-126 [29], a well-known EZH2 inhibitor, were added as internal controls. In NB4 cells, the levels of acetyl-H3 (acH3) and acetyl-H4 (acH4) as well as those of acetyl-H3K9/14 (AcH3K9/14, see Appendix A) and of the non-histone substrate acetyl-α-tubulin (Ac-Tubulin) were detected as an index of p300 inhibition (Figure 4). In the same cell line, the levels of H3K27me3, histone mark of the EZH2 catalytic activity (Figure 4), as well as the level of the EZH2 expression (see Appendix A) have been determined. Last, the amount of H3R17me2, the functional read-out of CARM1 inhibition, was determined (Figure 4).

Consistently with their IC_50_ values for p300 inhibition in enzyme assay (Table 1), the majority of the tested compounds showed a marked decrease of the levels of acetyl-H3 in NB4 cells with the only exception of the N1-phenylethyl- and -3-phenylpropyl-substituted **4f** and **4g**, which were less effective. A similar scenario was observed with the acetyl-H4, acetyl-H3K9/14, and acetyl-α-tubulin levels, which decreased with almost all the tested compounds, thus confirming their p300 inhibition in this cell context (Figure 4 and Appendix A).

Against H3K27me3, the mark of the catalytic activity of PRC2/EZH2, compounds **4d**–**j** showed no effects, according to their weak, if any, inhibition in enzyme assay (Table 1). Only compounds **4h** and **4m** displayed a reduction of the H3K27me3 levels (Figure 4). In addition, all the tested compounds reduced the level of the EZH2 expression in NB4 cells, confirming the behavior previously observed with the close analogue **3** in the same cell line [21] (Appendix A).

Regarding CARM1 inhibition, almost all the tested compounds decreased the levels of the H3R17me2 histone mark, with **4h**, **4l**, and **4m**, the last two potent at submicromolar level in Table 1, being the most effective (Figure 4).

In MCF-7 cells, the levels of acetyl-H3, acetyl-H4, H3K27me3, and H3R17me2 were detected by western blot analyses as functional tests for p300 (acH3 and acH4), EZH2 (H3K27me3) and CARM1 (H3R17me2) inhibition in cells, respectively (Figure 5). 

In acetyl-H3 assay all the compounds led to a consistent reduction of the levels of the acetylation mark, except for **4f** and **4g** (Figure 5). In acetyl-H4 and acetyl-α-tubulin assays, a limited number of the tested compounds (**4e**, **4i**, **4j**, **4l**, **4n** for acetyl-H4, and **4i** and **4k** for acetyl-α-tubulin) were able to give a decrease of the acetylation levels. About EZH2, few compounds (**4e**, **4h**, **4j**, **4m**, and **4n**) displayed a reduction of the H3K27me3 levels (Figure 5). Towards CARM1, **4i** and **4m** were the most effective in reducing the H3R17me2 levels (Figure 5).

## 3. Materials and Methods

### 3.1. Chemistry

Melting points were determined on a Buchi 530 melting point apparatus (Buchi Italia, Cornaredo, Italy). ^1^H-NMR and ^13^C-NMR spectra were recorded at 400 MHz using a Bruker AC 400 spectrometer (Bruker Italia, Milano, Italy); chemical shifts are reported in δ (ppm) units relative to the internal reference tetramethylsilane (Me_4_Si). Mass spectra were recorded on an API-TOF Mariner by Perspective Biosystem (Stratford, TX, USA), samples were injected by a Harvard pump using a flow rate of 5–10 μL/min, infused in the Electrospray system. All compounds were routinely checked by thin layer chromatography (TLC). TLC was performed on aluminum-backed silica gel plates (Merck DC, Alufolien Kieselgel 60 F254) with spots visualized by UV light (254 and 365 nm) or using a KMnO_4_ alkaline solution as staining reagent. All solvents were reagent grade and, when necessary, were purified and dried by standard methods. Concentration of solutions after reactions and extractions involved the use of a rotary evaporator operating at reduced pressure of ~20 Torr. Organic solutions were dried over anhydrous sodium sulfate (Na_2_SO_4_). Analytical results are within 0.40% of the theoretical values. All chemicals were purchased from Sigma Aldrich s.r.l. (Milan, Italy) or from TCI Europe N.V. (Zwijndrecht, Belgium), and were of the highest purity. As a rule, samples prepared for physical and biological studies were dried in high vacuum over P_2_O_5_ for 20 h at temperatures ranging from 25 to 40 °C, depending on the sample melting point.

#### 3.1.1. General Procedure for the Synthesis of the *N*-Alkylaryl 3,5-Bis((*E*)-2-bromobenzylidene)piperid-4-ones (**4e**–**i**). Example: 1-benzyl-3,5-bis((*E*)-2-bromobenzylidene)piperidin-4-one (**4e**)

3,5-Bis((*E*)-2-bromobenzyliden)piperidin-4-one **4c** [25] (0.46 mmol, 200 g, 1 eq.) and benzyl bromide (1.38 mmol, 164 µL, 3 eq.) were added to a suspension of anhydrous K_2_CO_3_ (0.69 mmol, 954 mg, 1.5 eq.) in acetonitrile (10 mL), and the resulting suspension was stirred at 60 °C. After 2 h the solvent was evaporated, water (50 mL) was added and the aqueous solution was extracted with dichloromethane (3 × 30 mL). The collected organic phases were washed with a saturated solution of NaCl (30 mL) and then dried with anhydrous Na_2_SO_4_, filtered and evaporated under vacuum to afford a crude that was purified on silica gel (AcOEt/*n*-hexane 1:3) to obtain the desired compound (170 mg, 67%) as a pale yellow solid. M.p.: 144–146 °C (toluene). Yield: 82%. ^1^H-NMR (DMSO-*d*_6_) δ ppm: 7.75–7.73 (m, 4H), 7.40 (t, 2H, *J* = 7.2 Hz), 7.34–7.30 (m, 4H), 7.17 (bs, 5H), 3.71 (bs, 4H), 3.60 (s, 2H). ^13^C-NMR (DMSO-*d*_6_) δ ppm: 161.4, 137.9 (2C), 135.3 (2C), 135.1 (2C), 134.7, 133.4 (2C), 131.4 (2C), 131.2 (2C), 129.2 (2C), 128.5 (2C), 128.2, 127.6 (2C), 124.9 (2C), 59.9, 53.3 (2C). MS (ESI-MS): Calculated: 524.00 for C_26_H_21_Br_2_NO [M + H]^+^, Found: 524.02. Calculated for M.W. 523.27, %: C 59.68; H 4.05; N 2.68; Br 30.54. Found, %: C 59.44; H 3.96; N 2.85; Br 30.68.

3,5-Bis((*E*)-2-bromobenzylidene)-1-phenethylpiperidin-4-one (**4f**). M.p.: 105–107 °C (cyclohexane). Yield: 65%. ^1^H-NMR (CDCl_3_) δ ppm: 7.98 (bs, 2H), 7.69 (d, 2H, *J* = 7.2 Hz), 7.36 (t, 2H, *J* = 7.2 Hz), 7.28–7.27 (m, 1H), 7.25–7.20 (m, 5H), 7.16 (d, 1H, *J* = 6.8 Hz), 7.09 (d, 2H, *J* = 7.6 Hz), 3.76 (bs, 4H), 2.75–2.73 (m, 2H), 2.70–2.65 (m, 2H). ^13^C-NMR (CDCl_3_) δ ppm: 182.6, 146.4 (2C), 140.0 (2C), 136.9, 135.2 (2C), 133.4 (2C), 129.9 (2C), 129.3 (2C), 128.9 (2C), 128.5 (2C), 127.0 (2C), 126.0, 125.1 (2C), 60.2, 52.4 (2C), 32.8. MS (ESI-MS): Calculated: 538.02 for C_27_H_24_Br_2_NO [M + H]^+^, Found: 538.03. Calculated for M.W. 537.30, %: C 60.36; H 4.31; N 2.61; Br 29.74. Found, %: C 60.59; H 4.39; N 2.42; Br 29.51.

3,5-Bis((*E*)-2-bromobenzylidene)-1-(3-phenylpropyl)piperidin-4-one (**4g**). M.p.: 93–95 °C (cyclohexane). Yield: 67%. ^1^H-NMR (CDCl_3_) δ ppm: 7.97 (bs, 2H), 7.68 (d, 2H, *J* = 8.0 Hz), 7.36 (t, 2H, *J* = 7.2 Hz), 7.29–7.26 (m, 2H), 7.25–7.20 (m, 4H), 7.16 (d, 1H, *J* = 6.8 Hz), 7.06 (d, 2H, *J* = 7.2 Hz), 3.68 (bs, 4H), 2.56 (t, 2H, *J* = 7.6 Hz), 2.49 (t, 2H, *J* = 7.6 Hz), 1.65 (q, 2H, *J* = 7.5 Hz). ^13^C-NMR (CDCl_3_) δ ppm: 182.5, 146.7 (2C), 140.3 (2C), 137.4, 135.1 (2C), 133.9 (2C), 129.5 (2C), 129.4 (2C), 128.9 (2C), 128.6 (2C), 127.7 (2C), 126.4, 125.0 (2C), 58.2, 53.6 (2C), 32.0, 29.8. MS (ESI-MS): Calculated: 552.04 for C_28_H_26_Br_2_NO [M + H]^+^, Found: 552.07. Calculated for M.W. 551.32, %: C 61.00; H 4.57; N 2.54; Br 28.99. Found, %: C 60.85; H 4.49; N 2.62; Br 29.11.

3,5-Bis((*E*)-2-bromobenzylidene)-1-(2-oxo-2-phenylethyl)piperidin-4-one (**4h**). M.p.: 109–111 °C (cyclohexane). Yield: 78%. ^1^H-NMR (CDCl_3_) δ ppm: 7.99 (bs, 2H), 7.88 (d, 2H, *J* = 7.2 Hz), 7.65 (d, 2H, *J* = 7.6 Hz), 7.57 (t, 1H, *J* = 7.6 Hz), 7.44 (t, 2H, *J* = 7.6 Hz), 7.34 (t, 2H, *J* = 7.6 Hz), 7.24–7.21 (m, 4H), 4.01 (s, 2H), 3.98 (bs, 4H). ^13^C-NMR (DMSO-*d*_6_) δ ppm: 197.3, 186.6, 136.0 (2C), 135.2 (2C), 134.7, 134.7 (2C), 133.8, 133.4 (2C), 131.4 (2C), 131.3 (2C), 129.1 (2C), 128.3 (2C), 128.3 (2C), 125.0 (2C), 62.2, 53.9 (2C). MS (ESI-MS): Calculated: 551.99 for C_27_H_21_Br_2_NO_2_ [M + H]^+^, Found: 552.00. Calculated for M.W. 551.28, %: C 58.83; H 3.84; N 2.54; Br 28.99. Found, %: C 58.99; H 3.91; N 2.32; Br 28.79.

3,5-Bis((*E*)-2-bromobenzylidene)-1-(3-oxo-3-phenylpropyl)piperidin-4-one (**4i**). M.p.: 95–97 °C (cyclohexane). Yield: 69%. 7.89 (d, 2H, *J* = 7.6 Hz), 7.75 (bs, 2H), 7.56–7.51 (m, 4H), 7.44 (t, 2H, *J* = 7.6 Hz), 7.32–7.31 (m, 3H), 7.29 (s, 2H), 3.88 (s, 4H), 3.13 (t, 2H, *J* = 6.0 Hz), 2.60 (t, 2H, *J* = 6.4 Hz). ^13^C-NMR (CDCl_3_) δ ppm: 196.4, 182.1, 144.8 (2C), 140.5 (2C), 137.2, 135.5 (2C), 134.0, 132.3 (2C), 130.5 (2C), 129.3 (2C), 128.5 (2C), 127.3 (2C), 127.0 (2C), 125.0 (2C), 59.3, 52.4 (2C), 40.7. MS (ESI-MS): Calculated: 566.01 for C_28_H_23_Br_2_NO_2_ [M + H]^+^, Found: 566.02. Calculated for M.W. 565.31, %: C 59.49; H 4.10; N 2.48; Br 28.27. Found, %: C 59.27; H 3.96; N 2.61; Br 28.45.

#### 3.1.2. Synthesis of 3,5-Bis((*E*)-2-bromobenzylidene)-1-(4-oxo-4-phenylbutyl)piperidin-4-one (**4j**)

To a stirring solution of the 3,5-bis((*E*)-2-bromobenzylidene)piperidin-4-one (**4c**) [25] (0.46 mmol, 200 mg) and 4-oxo-4-phenylbutanal (0.46 mmol, 75 mg) in anhydrous dichloromethane (10 mL), sodium triacetoxyborohydride (0.55 mmol, 117 mg, 1.2 eq.) was added portion wise and the resulting suspension was stirred at room temperature. After 2 h the reaction was quenched with H_2_O (30 mL) and extracted with dichloromethane (4 × 20 mL). The collected organic phases were then washed with a NaCl saturated aqueous solution (30 mL) and then dried with anhydrous Na_2_SO_4_, filtered and evaporated under vacuum to afford a crude that was purified on silica gel (AcOEt/*n*-hexane 1:3) to obtain the desired compound (197 mg, 74%). M.p.: 84–86 °C (cyclohexane). ^1^H-NMR (CDCl_3_) δ ppm: 7.90 (d, 2H, *J* = 7.6 Hz), 7.72 bs, 2H), 7.54–7.52 (m, 5H), 7.42 (t, 2H, *J* = 8 Hz), 7.33–7.32 (m, 4H), 3.80 (s, 4H), 2.99 (t, 2H, *J* = 7.2 Hz), 2.67 (t, 2H, *J* = 6.4 Hz), 1.94–1.90 (q, 2H, *J* = 7.2 Hz). ^13^C-NMR (CDCl_3_) δ ppm: 196.2, 182.3, 144.2 (2C), 141.0 (2C), 137.3, 135.5 (2C), 134.4, 132.1 (2C), 129.5 (2C), 128.3 (2C), 128.1 (2C), 127.7 (2C), 127.5 (2C), 125.1 (2C), 57.3, 52.5 (2C), 37.7, 25.0. MS (ESI-MS): Calculated: 580.02 for C_29_H_25_Br_2_NO_2_ [M + H]^+^, Found: 580.01. Calculated for M.W. 579.33, %: C 60.12; H 4.35; N 2.42; Br 27.58. Found, %: C 60.32; H 4.45; N 2.29; Br 27.34.

#### 3.1.3. General Procedure for the Synthesis of *N*-Acyl-3,5-Bis((*E*)-2-bromobenzylidene)piperidin-4-ones (**4k**–**n**). Example: 3,5-Bis((*E*)-2-bromobenzylidene)-1-cinnamoylpiperidin-4-one (**4n**)

Cinnamoyl chloride (0.67 mmol, 100 µL) was slowly added at 0 °C to a stirring solution of **4c** [25] (0.45 mmol, 195 mg) and Et_3_N (0.76 mmol, 110 µL, 1.13 eq.) in anhydrous dichloromethane (5 mL). The resulting mixture is then allowed to stir at room temperature. After 1 h the reaction was quenched with water (50 mL) and extracted with dichloromethane (3 × 30 mL). The collected organic layers were washed with 2N HCl (3 × 30 mL) and then with a saturated solution of NaCl (30 mL). The organic phase was dried with anhydrous Na_2_SO_4_, filtered and evaporated under vacuum to afford a crude that was then purified on silica gel (AcOEt/*n*-hexane 1:3) to afford the desired product (152 mg, 60%). M.p.: 98–100 °C (cyclohexane). ^1^H-NMR (CDCl_3_) δ ppm: 7.95 (bs, 2H), 7.71 (d, 2H, *J* = 8 Hz), 7.54 (d, 2H, *J* = 15.6 Hz), 7.44–7.40 (m, 3H), 7.39–7.31 (m, 5H), 7.20 (d, 2H, *J* = 6.8 Hz), 6.38 (d, 1H, *J* = 15.6 Hz), 4.80–4.78 (m, 4H). ^13^C-NMR (CDCl_3_) δ ppm: 185.6, 165.3, 145.5 (2C), 143.5, 141.9 (2C), 137.0, 135.1 (2C), 133.6 (2C), 129.2 (2C), 128.7 (2C), 128.3, 127.9 (2C), 127.6 (2C), 126.0 (2C), 125.6 (2C), 117.1, 48.6 (2C). MS (ESI-MS): Calculated: 563.99 for C_28_H_21_Br_2_NO_2_ [M + H]^+^, Found: 564.01. Calculated for M.W. 563.29; %: C 59.70; H 3.76; N 2.49; Br 28.37. Found, %: C 59.49; H 3.65; N 2.70; Br 28.54.

1-Benzoyl-3,5-bis((*E*)-2-bromobenzylidene)piperidin-4-one (**4k**). M.p.: 124–126 °C (cyclohexane/toluene). Yield: 76%. ^1^H-NMR (CDCl_3_) δ ppm: 8.01 (s, 2H), 7.69 (d, 2H, *J* = 8.8 Hz), 7.23–7.21 (3H, m), 7.14–7.11 (8H, m), 4.75–4.45 (m, 4H). ^13^C-NMR (CDCl_3_) δ ppm: 182.5, 169.1, 144.8 (2C), 139.8 (2C), 136.1, 133.8 (2C), 132.1 (2C), 128.9, 128.5 (2C), 128.3 (2C), 127.8 (2C), 126.1 (2C), 125.8 (2C), 124.7 (2C), 50.1 (2C). MS (ESI-MS): Calculated: 537.98 for C_26_H_19_Br_2_NO_2_ [M + H]^+^, Found: 538.00. Calculated for M.W. 537.25, %: C 58.13; H 3.56; N 2.61; Br 29.75. Found, %: C 58.32; H 3.68; N 2.49; Br 29.67.

3,5-Bis((*E*)-2-bromobenzylidene)-1-(2-phenylacetyl)piperidin-4-one (**4l**). M.p.: 126–128 °C (cyclohexane/benzene). Yield: 67%. ^1^H-NMR (CDCl_3_) δ ppm: 7.95 (d, 2H, *J* = 18.8 Hz), 7.76 (d, 1H, *J* = 6.8 Hz), 7.68 (d, 1H, *J* = 7.6 Hz), 7.45–7.36 (m, 4H), 7.26–7.24 (m, 1H), 7.20–7.18 (m, 3H), 7.08 (d, 1H, *J* = 6.8 Hz), 6.84–6.82 (m, 2H), 4.77 (bs, 2H), 4.45 (bs, 2H), 3.50 (s, 2H). ^13^C-NMR (DMSO-*d*_6_) δ ppm: 186.1, 169.8, 135.2 (2C), 134.4, 134.0, 133.6 (4C), 131.8 (2C), 131.4 (2C), 128.8 (5C), 128.4 (2C), 127.0 (2C), 125.5 (1C), 47.8, 44.5, 42.8. MS (ESI-MS): Calculated: 551.99 for C_27_H_21_Br_2_NO_2_ [M + H]^+^, Found: 552.00. Calculated for M.W. 551.28, %: C 58.83; H 3.84; N 2.54; Br 28.99. Found, %: C 59.04; H 3.97; N 2.35; Br 28.75.

3,5-Bis((*E*)-2-bromobenzylidene)-1-(3-phenylpropanoyl)piperidin-4-one (**4m**). M.p.: 95–97 °C (cyclohexane). Yield: 69%. ^1^H-NMR (CDCl_3_) δ ppm: 7.94 (d, 2H, *J* = 18.8 Hz), 7.68 (d, 2H, *J* = 8.0 Hz), 7.45–7.39 (m, 3H), 7.25–7.21 (m, 4H), 7.19–7.14 (m, 2H), 6.98 (d, 2H, *J* = 7.2 Hz), 4.76 (bs, 2H), 4.41 (bs, 2H), 2.83 (t, 2H, *J* = 8.0 Hz), 2.39 (t, 2H, *J* = 8.0 Hz). ^13^C-NMR (CDCl_3_) δ ppm: 182.3, 168.4, 144.3 (2C), 142.1, 140.1 (2C), 134.1 (2C), 133.0 (2C), 128.6 (2C), 128.3 (2C), 127.3, 128.0 (2C), 126.6 (2C), 125.0 (2C), 124.7 (2C), 48.5 (2C), 35.7, 32.9. MS (ESI-MS): Calculated: 566.01 for C_28_H_23_Br_2_NO_2_ [M + H]^+^, Found: 566.01. Calculated for M.W. 565.31, %: C 59.49; H 4.10; N 2.48; Br 28.27. Found, %: C 59.29; H 3.97; N 2.72; Br 28.45.

### 3.2. Biochemistry

#### 3.2.1. HAT Assays

The effect of tested derivatives on the catalytic activity of p300 and PCAF was determined in a HotSpot HAT activity assay by Reaction Biology Corporation (Malvern, PA, USA) according to the company’s standard operating procedure. In brief, the recombinant catalytic domains of PCAF (aa 492–658, 250 nM) or p300 (aa 1284–1673, 30 nM) were incubated with histone H3 as substrate (5 µM) and [Acetyl-^3^H]-Acetyl Coenzyme A (3 µM) as an acetyl donor in reaction buffer (50 mM Tris-HCl (pH 8.0), 50 mM NaCl, 0.1 mM EDTA, 1 mM DTT, 1 mM PMSF, 1% DMSO) for 1 h at 30 °C in the presence or absence of a dose titration of the compounds. The reaction mixture was delivered to filter-paper for detection. IC_50_ values were analyzed using Excel and GraphPad Prism 6.0 software (GraphPad Software Inc., San Diego, CA, USA). The experiments for p300 were performed in triplicate.

#### 3.2.2. SIRT1/2 Assay

The SIRT inhibition assay was performed using human recombinant SIRT1 and SIRT2 produced in *E. coli*. Compounds were tested using a modified Fluor de Lys fluorescence-based assay kit based on the method described in the BioMol product sheet (AK-555, AK-556, Milan, Italy). The assay procedure has two steps: in the first part the SIRT1/2 substrate, an acetylated Lys side chain comprising amino acids 379-382 (Arg-His-Lys-Lys(Ac)) (BioMol KI177, Milan, Italy) of human p53 conjugated with aminomethylcoumarin is deacetylated during incubation at 37 °C for 1 h by SIRT1 or SIRT2 in the presence of NAD^+^ and the tested compounds. The second stage is initiated by the addition of the Developer II (KI176), including nicotinamide (NAM), nicotinamide that stops the SIRT1/2 activity, and the fluorescent signal is produced. The fluorescence was measured on a fluorometric reader (Infinite 200 TECAN, Männedorf, Switzerland) with excitation set at 360 nm and emission detection set at 460 nm.

#### 3.2.3. EZH2/PRC2 Assay

The EZH2 substrate (0.05 mg/mL core histone) was added in the freshly prepared reaction buffer (50 mM Tris-HCl (pH 8.0), 50 mM NaCl, 1 mM EDTA, 1 mM DTT, 1 mM PMSF, 1% DMSO). The PRC2 complex (complex of human EZH2, human EED, human SUZ12, human AEBP2, and human RbAp48) was delivered into the substrate solution and the mixture was mixed gently. Afterwards, the tested compounds dissolved in DMSO were delivered into the enzyme/substrate reaction mixture by using Acoustic Technology (Echo 550, LabCyte Inc. Sunnyvale, CA, USA) in nanoliter range, and ^3^H-SAM was added into the reaction mixture to initiate the reaction. The reaction mixture was incubated for 1 h at 30 °C and then it was delivered to filter-paper for detection. The data were analyzed using Excel and GraphPad Prism software for IC_50_ curve fits. The experiments were performed in duplicate.

#### 3.2.4. CARM1 Assay

Histone H3 (5 μM) was added in freshly prepared reaction buffer (50 mM Tris-HCl (pH 8.5), 5 mM MgCl_2_, 50 mM NaCl, 0.01% Brij35, 1 mM DTT, 1% DMSO). CARM1 was delivered into the substrate solution and the mixture was mixed gently. Afterwards, the tested compounds dissolved in DMSO were delivered into the enzyme/substrate reaction mixture by using Acoustic Technology (Echo 550, LabCyte Inc. Sunnyvale, CA) in nanoliter range, and 1 µM ^3^H-SAM was also added into the reaction mixture to initiate the reaction. The reaction mixture was incubated for 1 h at 30 °C and then it was delivered to filter-paper for detection. The data were analyzed using Excel and GraphPad Prism software for IC_50_ curve fits. The experiments were performed in triplicate.

### 3.3. Cellular Studies

#### 3.3.1. Cell Cultures

NB4 tumor cell line was purchased by DSMZ, Braunschweig Germany, and U937, NB4, MCF7, and SH-SY5Y tumor cell lines by American Type Culture Collection (ATCC, Milan, Italy). Cell lines have been tested and authenticated following manufacturer’s instruction. All cell lines were maintained in an incubator at 37 °C and 5% CO_2_. The human leukemia cells were grown in RPMI-1640 (Sigma-Aldrich, Milan, Italy) while human breast cancer and neuroblastoma cells in Dulbecco’s Modified Eagle Medium (DMEM) (Sigma-Aldrich) culture media, 1% l-glutamine (EuroClone, Milan, Italy), 10% heat-inactivated Fetal Bovine Serum (FBS) (GIBCO, Monza, Italy) and antibiotics. MePR2B cells were grown as previously reported [26].

#### 3.3.2. Cell Proliferation, Cell Cycle, and Cell Death Analyses

For colorimetric exclusion, the cells (2 × 10^5^ cells/mL) were plated in multiwells in triplicate. After stimulations at different times and concentrations, cells were diluted in the ratio 1:1 in Trypan Blue (Sigma) and counted with an optical microscope. For cell cycle analyses, cells were plated (2 × 10^5^ cells/mL) and were harvested, centrifuged at 1200 rpm for 5 min and resuspended in 500 μL of a hypotonic solution containing 1× PBS, Sodium Citrate 0.1%, 0.1% NP-40, RNAase A and 50 mg/mL Propidium Iodide (PI). After 30 min at room temperature (RT) in the dark, samples were acquired by FACS-Calibur (BD Bioscences, San Jose, CA, USA) using CellQuest software (BD Biosciences). The percentage in different phases of the cell cycle was determined by ModFit LT V3 software (Verity). All experiments were performed in triplicate. Cell death was measured as percentage of cells in sub-G1 phase as in [30].

#### 3.3.3. Western Blot Analyses

The procedures were performed as described in [30,31]. Primary antibodies used were: Ac-H3 (Millipore, Milan, Italy); Ac-H4 (Millipore, Milan, Italy); Ac-H3K9-14 (Diagenode, Ougrée, Belgium); Ac-α-tubulin (Sigma-Aldrich), H3K27me3 (Diagenode); H3R17me2 (Diagenode); EZH2 (Abcam, Cambridge, UK). H3 (Sigma-Aldrich), H4 (Abcam), GAPDH (Sigma) and ERKs (Santa Cruz, Biotechnology, TX, USA) were used to normalize the total protein extracts (see also Appendix A).

## 4. Conclusions

Curcuminoids are a wide family of variegated compounds bearing a plethora of therapeutic activities, including anti-inflammatory, antidiabetic, neuroprotective, and anticancer activities [24,25,32,33,34,35,36,37,38]. Some related compounds reported by us as epi-MLs exerted biochemical inhibition of a number of epigenetic targets (PRMTs, KMTs, p300, SIRT1/2), joined to potent induction of apoptosis and/or cytodifferentiation in U937 AML cells [13]. The changes at decoration of the two phenyl rings as well as the use of different spacers connecting them were able to focus the activity of such compounds against specific epigenetics targets, leading to more selective derivatives [12,14,17]. In particular, the 1-benzyl-3,5-bis((*E*)-3-bromobenzylidene)piperidin-4-one **3** behaved as dual p300/EZH2 inhibitor and displayed cancer-selective growth arrest and cell death in hematological malignancies as well as solid tumors, in in vitro, ex vivo, and in vivo models [21,22].

Starting from these data, we decided to explore the effect of the shift of the bromine substitution at the two phenyl ring from the 3 to the 2 position, using as connection unit cyclohexanone, tetrahydro-4*H*-pyran-4-one or piperidin-4-one moiety (compounds **4a**–**n**). Such compounds were evaluated in p300, PCAF, SIRT1/2, EZH2 and CARM1 enzyme assays, and in NB4 APL, U937 AML, MCF-7 breast cancer and SH-SY5Y neuroblastoma cells, to determine their effects in cell cycle phases and apoptosis induction when used at 5 μM for 30 h. Generally, in biochemical assays **4a**–**n** exhibited high inhibitory activities against p300 and CARM1, weak activity (if any) against PRC2/EZH2, whereas were totally inactive against PCAF and SIRT1/2. In particular, compounds carrying the cyclohexanone (**4a**), tetrahydro-4*H*-pyran-4-one (**4b**), and piperidin-4-one substituted at N1 with a methyl, benzyl, oxophenylalkyl and acyl moieties (**4d**–**n**) displayed the highest inhibition, with **4l** and **4m** reaching submicromolar IC_50_ values for both p300 and CARM1 enzymes. In cellular assays, the majority of **4a**–**n** gave an alteration of the cell cycle with pro-apoptotic effect in the tested cell lines, with the leukemia cells being generally more sensitive than the solid cancer cells. In NB4 cells, **4l** and **4m** induced the strongest apoptosis (near 70%). In U937 cells, in addition to **4l** and **4m** also the N1 benzyl (**4e**) and benzoyl (**4k**) analogues exhibited near or over 40% apoptosis induction. In MCF-7 and in SH-SY5Y cells the effect is lower: the N1 2-oxo-2-phenylethyl **4h** and the N1 acyl derivatives **4k**–**m** were the most potent but the induction of apoptosis was around 25% (MCF-7) or less than 20% (SH-SY5Y) at the tested conditions. Despite some apparent discrepancies, due to the differences between the in vitro enzymatic assays and the more complex cellular system, Western blot analysis of selected derivatives in NB4 and MCF-7 cells, performed to detect the levels of acetyl-H3/acetyl-H4/acetyl-α-tubulin, H3K27me3 and H3R17me2 as histone marks for cellular activity of p300, EZH2, and CARM1, respectively, supports the role of the recruitment and inhibition of these epigenetic targets, mainly p300 and CARM1, in the mechanism of anticancer activities of the studied compounds.

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
