# Peer review of "Properly Substituted Cyclic Bis-(2-bromobenzylidene) Compounds Behaved as Dual p300/CARM1 Inhibitors and Induced Apoptosis in Cancer Cells"

_molecules, 2020, doi:10.3390/molecules25143122_

Round 1

Reviewer 1 Report

In this paper Author reported a series of bis((E)-2-bromobenzylidene) cyclic compounds to test in biochemical (p300, PCAF, EZH2 and CARM1) and cellular (NB4, U937, MCF-7, SH-SY5Y) assays. The majority of synthesized compounds exhibited potent dual p300 and CARM1 inhibition, sometimes reaching the submicromolar level, and induction of apoptosis mainly in the tested leukemia cell lines. Western blot analyses performed in NB4 and MCF-7 cells on selected compounds confirmed their inhibition of p300 and CARM1 through decrease of the levels of acetyl-H3 and acetyl-H4, marks for p300 inhibition, and of H3R17me2, mark for CARM1inhibition.

The paper is of good scientific interest, well written, clear and interesting.

In my opinion paper can be accepted with minor revisions. In detail:

  • The introduction is too long and both the part relating to polypharmacology and the part relating to the previous synthesized compounds should be shortened, inserting only compounds very similar to the newly synthesized compounds 7 (in my opinion compounds 1-3 could be removed).
  • Consequently in Figure 1 biological activity of previous compounds should be removed, because these informations have been reported in the introduction part.
  • Figure 1: compounds 7a-n “alkoxyaryl. acyl” should be changed in “alkoxyaryl, acyl”, comma insetad of point.
  • the experimental procedures of 7a, 7b, 7c and 7d should be delete if the synthetic methods are the same of literature (references 24-26).
  • Consequently in scheme 1 synthesis of 7a-d should be delete.
  • Scheme 1: please insert reaction times and yields.

Author Response

In this paper Author reported a series of bis((E)-2-bromobenzylidene) cyclic compounds to test in biochemical (p300, PCAF, EZH2 and CARM1) and cellular (NB4, U937, MCF-7, SH-SY5Y) assays. The majority of synthesized compounds exhibited potent dual p300 and CARM1 inhibition, sometimes reaching the submicromolar level, and induction of apoptosis mainly in the tested leukemia cell lines. Western blot analyses performed in NB4 and MCF-7 cells on selected compounds confirmed their inhibition of p300 and CARM1 through decrease of the levels of acetyl-H3 and acetyl-H4, marks for p300 inhibition, and of H3R17me2, mark for CARM1inhibition.

The paper is of good scientific interest, well written, clear and interesting.

In my opinion paper can be accepted with minor revisions. In detail:

  • The introduction is too long and both the part relating to polypharmacology and the part relating to the previous synthesized compounds should be shortened, inserting only compounds very similar to the newly synthesized compounds 7(in my opinion compounds 1-3 could be removed).

We thank a lot the reviewer for this very useful suggestion. We reduced both the polypharmacology and the previous researches parts, removing compounds 1-3 and renumbering all the others.

  • Consequently in Figure 1 biological activity of previous compounds should be removed, because these informations have been reported in the introduction part.

Figure 1 has been modified accordingly to the referee’s suggestion.

  • Figure 1: compounds 7a-n“alkoxyaryl. acyl” should be changed in “alkoxyaryl, acyl”, comma insetad of point.
  •  
  • the experimental procedures of 7a7b7cand 7d should be delete if the synthetic methods are the same of literature (references 24-26).

The procedures for 7a-d have been removed.

  • Consequently in scheme 1 synthesis of 7a-d should be delete.

Done.

  • Scheme 1: please insert reaction times and yields.

Done.

Reviewer 2 Report

In this manuscript, the authors provide a rationale for the discovery of compounds that target multiple epigenetic enzymes, building upon their past work in this area.  In the work at hand, a select group of compounds bearing  2'-bromobenzylidine groups is prepared and assayed against two HAT enzymes and two methyltransferase enzymes.  Cell cycle analysis and Western blot results are also presented.  The chemistry is straightforward, and some of the compounds reported here have previously been reported in the literature. The biological characterization of the compounds is problematic.  However, if major revisions are made, the manuscript may provide some solid additional SAR in this area. 

The most serious issue with the manuscript is with the biological data.  The  enzyme inhibition assays are not interpretable, as there were no replicates run and therefore there is no estimate of error for the results.  The authors should first conduct cytotoxicity studies on the compounds prior to the cell cycle and apoptosis analysis. As it is, the assay concentration of 5 µM for each compound is not meaningful.  There are problems with the Western blots: In one gel SAHA is clearly causing an increase in AcH3 in NB4 cell, where as in the adjacent gel there does not appear to be a difference between the SAHA-treated cells and control.  More globally - there is no attempt to quantify these gels and, again there are no replicates. The loading standards ERK and ponceau red do not appear to be appropriate for these gels. In the case of ERK, there is no assurance the compound treatment does not affect levels, and the signal for ponceau red may be too weak to allow quantification.  The authors should include as positive control a known KMT inhibitor.

Although the chemistry is straight-forward, there are also issues with this portion of the work. Authors should indicate which of these compounds have been previously reported in the literature (7a-e) with citations to these prior syntheses. There is no way to assess the purity of the compounds given the data provided.  The authors should at least provide copies of the 1H and 13C spectra for these compounds in supporting information.

In addition to these major points, there are a few more minor issues the authors should also address.  Despite clearly showing in the introduction the cross-inhibition of a wide range of epigenetic targets, the authors provide data here for relatively few targets: the methyltransferases CARM1 (PRMT4) and PRC2, and two HATs (p300 and PCAF). Can the authors also provide data for additional classes (e.g., sirtuin)? On page 4, line 148: "..exhibited somehow activity..." should be "some".  On page 5, the incubation time on line 190 is 24 h, but in the Figure 4 legend is it 30 h. On page 5-6, Figures 2 and 3, the data for compound 7n is missing.

Author Response

In this manuscript, the authors provide a rationale for the discovery of compounds that target multiple epigenetic enzymes, building upon their past work in this area.  In the work at hand, a select group of compounds bearing  2'-bromobenzylidine groups is prepared and assayed against two HAT enzymes and two methyltransferase enzymes.  Cell cycle analysis and Western blot results are also presented.  The chemistry is straightforward, and some of the compounds reported here have previously been reported in the literature. The biological characterization of the compounds is problematic.  However, if major revisions are made, the manuscript may provide some solid additional SAR in this area. 

  • The most serious issue with the manuscript is with the biological data.  The  enzyme inhibition assays are not interpretable, as there were no replicates run and therefore there is no estimate of error for the results.  

We repeated the experiments for p300 and CARM1, now in triplicate, and for EZH2, now in duplicate. The relative standard deviations for the enzyme data have been determined. All the new data have been included in Table 1, together with inhibiting data against SIRT1 and SIRT2, as request below.

  • The authors should first conduct cytotoxicity studies on the compounds prior to the cell cycle and apoptosis analysis. As it is, the assay concentration of 5 µM for each compound is not meaningful.  

We performed MTT studies on non-cancer MePR2B cells with 7l and 7m (now 4l and 4m), the most potent compounds of the series as apoptosis inducers, and the relative effects on cell viability at 5 µM for 30 h have been reported in the main text (see page 11 of the revised version).

  • There are problems with the Western blots: In one gel SAHA is clearly causing an increase in AcH3 in NB4 cell, where as in the adjacent gel there does not appear to be a difference between the SAHA-treated cells and control.  More globally - there is no attempt to quantify these gels and, again there are no replicates. The loading standards ERK and ponceau red do not appear to be appropriate for these gels. In the case of ERK, there is no assurance the compound treatment does not affect levels, and the signal for ponceau red may be too weak to allow quantification.  The authors should include as positive control a known KMT inhibitor.

We repeated the WB for acetyl-H3, acetyl-H4, H3K27me3 and H3R17me2 in NB4 and MCF-7 cells treated with 4d-n, and we inserted the new gels to the manuscript in Figures 4 and 5. H3, H4 and GAPDH were used for equal loading. In addition to SAHA, GSK-126 as a well-known EZH2 inhibitor was added to the assays.

  • Although the chemistry is straight-forward, there are also issues with this portion of the work. Authors should indicate which of these compounds have been previously reported in the literature (7a-e) with citations to these prior syntheses.

As suggested by reviewer 1, we removed the characterization data for 4a-d and left only the relative references.

  • There is no way to assess the purity of the compounds given the data provided.  The authors should at least provide copies of the 1H and 13C spectra for these compounds in supporting information.

Copies of the 1H and representative 13C spectra have been added in Supplementary material.

  • In addition to these major points, there are a few more minor issues the authors should also address.  Despite clearly showing in the introduction the cross-inhibition of a wide range of epigenetic targets, the authors provide data here for relatively few targets: the methyltransferases CARM1 (PRMT4) and PRC2, and two HATs (p300 and PCAF). Can the authors also provide data for additional classes (e.g., sirtuin)?

The inhibition data for SIRT1 and SIRT2 have been added in Table 1. The tested compounds are totally inactive up to 200 μM.

  • On page 4, line 148: "..exhibited somehow activity..." should be "some".  

The required changes have been done.

  • On page 5, the incubation time on line 190 is 24 h, but in the Figure 4 legend is it 30 h.

The incubation time was 30 h, we apologize for the typo error.

  • On page 5-6, Figures 2 and 3, the data for compound 7n is missing.

Figure 2 and 3 must be made smaller in the Molecules format making impossible to see the data of 7n. In the revised version we increased the size of Figures 2 and 3 so that data for 4n are completely visible.

Round 2

Reviewer 2 Report

In this revised manuscript, the authors have addressed the issues raised in the review review of the original manuscript.  However, there are two minor points that should be addressed:

Carbonyl is missing in the general structure for 4k-n in Scheme 1 bottom right.

In general terms, the correlation between biochemical inhibition results and the Western blot results are not as clear as the authors claim.  For example, the authors need to discuss Western blot result of 4k, which is on par with 4g in terms of poor inhibition of acetyl marks AcH4 and tubulin, yet unlike 4g, is active in decreasing acH3 marks. Similarly, 4k and 4i are not the most potent compounds in the p300 inhibition assay, but are clearly better at decreasing tubulin acetyl marks in MCF-7 cells.  

The bottom line is that the data is a little more complicated than the authors conclude, and the final statement in the conclusion section should be changed to reflect that "Western blot analysis...supports the role of recruitment and inhibition of... p300 and CARM1, in the mechanism.." 

Author Response

  • Carbonyl is missing in the general structure for 4k-n in Scheme 1 bottom right.

The reviewer is absolutely right, we apologize a lot for the mistake, that was promptly corrected.

  • In general terms, the correlation between biochemical inhibition results and the Western blot results are not as clear as the authors claim.  For example, the authors need to discuss Western blot result of 4k, which is on par with 4g in terms of poor inhibition of acetyl marks AcH4 and tubulin, yet unlike 4g, is active in decreasing acH3 marks. Similarly, 4k and 4i are not the most potent compounds in the p300 inhibition assay, but are clearly better at decreasing tubulin acetyl marks in MCF-7 cells.
  • The bottom line is that the data is a little more complicated than the authors conclude, and the final statement in the conclusion section should be changed to reflect that "Western blot analysis...supports the role of recruitment and inhibition of... p300 and CARM1, in the mechanism.." 

Also in this case we totally agree with the reviewer: the match between enzyme and cellular (WB) data can be unclear sometimes, due to the great difference between the two systems. We included this warning in the main text and modified the conclusion as suggested.